# MEMS-tunable dielectric metasurface lens

Ehsan Arbabi [1], Amir Arbabi[1,2], Seyedeh Mahsa Kamali[1], Yu Horie [1], MohammadSadegh Faraji-Dana[1] & Andrei Faraon [1]

Varifocal lenses, conventionally implemented by changing the axial distance between multiple optical elements, have a wide range of applications in imaging and optical beam scanning. The use of conventional bulky refractive elements makes these varifocal lenses large, slow, and limits their tunability. Metasurfaces, a new category of lithographically defined diffractive devices, enable thin and lightweight optical elements with precisely engineered phase profiles. Here we demonstrate tunable metasurface doublets, based on microelectromechanical systems (MEMS), with more than 60 diopters (about 4%) change in the optical power upon a 1-μm movement of one metasurface, and a scanning frequency that can potentially reach a few kHz. They can also be integrated with a third metasurface to make compact microscopes (~1 mm thick) with a large corrected field of view (~500 μm or 40 degrees) and fast axial scanning for 3D imaging. This paves the way towards MEMS-integrated metasurfaces as a platform for tunable and reconfigurable optics.

[1] T. J. Watson Laboratory of Applied Physics, California Institute of Technology, 1200 E. California Blvd., Pasadena, CA 91125, USA. [2] Department of Electrical and Computer Engineering, University of Massachusetts Amherst, 151 Holdsworth Way, Amherst, MA 01003, USA. Correspondence and requests for materials should be addressed to A.F. (email: faraon@caltech.edu)

Lenses are ubiquitous optical elements present in almost all imaging systems. Compact lenses with tunable focal/imaging distance have many applications, and therefore several methods have been developed to make such devices[1–18]. Deformable solid and liquid-filled lenses with mechanical[1], electromechanical[2,12], electrowetting[9,10], and thermal[11] tuning mechanisms have been demonstrated. Although these devices are more compact than regular multi-element varifocal lenses, they are still bulky (since they are regular refractive devices), and have low tuning speeds (ranging from a few Hz to a few tens of Hz). Liquid crystal lenses with tunable focus[3–5] have higher tuning speeds, but they suffer from polarization dependence and limited tuning range. Freeform optical elements (e.g., Alvarez lenses) that can tune the focal distance upon lateral displacement of the elements have also been demonstrated[13,14]. These devices are generally based on mechanical movement of bulky elements and are therefore not very compact nor fast. Highly tunable diffractive and metasurface lenses based on stretchable substrates[15,16,19] have also been demonstrated, but they have low speeds and require a radial stretching mechanism that might increase the total device size. Spatial light modulators (SLMs) and other types of diffractive elements that have pixels with controllable phase shifts have been used and proposed[17,18] to achieve tunable beam steering and focusing. Liquid crystal based SLMs are polarization dependent and have limited speed and numerical apertures, and other proposals yet await an experimental demonstration of phase tuning over two dimensional arrays with high efficiency[17,20].

Optical metasurfaces[21,22] are planar arrangements of scatterers designed to manipulate various properties of an incident light beam. In the optical regime, dielectric metasurfaces are very versatile as they allow for wavefront control with high efficiency and subwavelength resolution. Several devices with the ability to control the phase[23–37], polarization[38,39], polarization and phase[40], or spectral components of light through harmonic generation[41–43] or filtering[44–47] have been demonstrated. Their thin form factor makes them suitable for development of ultrathin conformal optical elements[48,49], and their compatibility with conventional micorfabrication techniques allows for monolithic fabrication of optical systems consisting of multiple metasurfaces on a single chip[50,51]. These characteristics (i.e., the ability to precisely control the phase with subwavelength resolution and high gradients, thin and light form factor, and compatibility with microfabrication techniques) also make them very attractive for integration with the microelectromechanical systems (MEMS) technology to develop metasurface-based micro-opto-electromechanical systems (MOEMS). To date, integration of metasurfaces and MEMS devices has been limited to moving uniform high-contrast grating mirrors to tune the resonance wavelength of Fabry-Perot cavities[52,53], or change roundtrip propagation length of light to form spatial light modulators[54].

In this manuscript, we propose and demonstrate a metasurface doublet composed of a converging and a diverging metasurface lens with an electrically tunable focal distance. The large and opposite-sign optical powers of the two elements, as well as their very close proximity, make it possible to achieve large tuning of the optical power (~60 diopters, corresponding to about 4%) with small movements of one element (~1 micron). We have developed a fabrication process for making such metasurface doublets, and experimentally show metasurface lenses with over 60 μm tuning of the effective focal length (EFL) from 565 to 629 μm, corresponding to a ~180-diopter change in the optical power. Arrays of these devices can be fabricated on the same chip to allow for multiple lenses with different focal distances scanning different depths with frequencies potentially reaching several kHz. In addition, we show that such devices can be combined with the recently demonstrated monolithic metasurface optical

systems design[50] to develop compact focus-scanning objectives with corrected monochromatic aberrations over a large field of view. It is worth noting that MOEMS devices with the ability to axially scan the focus have previously been demonstrated based on integration of refractive and Fresnel microlenses with axially moving frames[6–8,55]. However, in these devices the focal point is scanned by the same distance that the lens is moved, and the effective focal length (or equivalently the optical power) is not actually tuned. Nevertheless, the concepts and techniques used in such devices can be combined with the metasurface doublet demonstrated here to achieve enhanced functionalities (e.g., enable lateral scanning of focus).

## Results

**Concept and design.** Figure 1a shows a schematic of the tunable focus doublet. The system consists of a stationary metasurface on a glass substrate, and a moving metasurface on a $SiN_x$ membrane. The membrane can be electrostatically actuated to change the distance between the two metasurfaces. The lenses are designed such that a small change in the distance between them, $\Delta x \sim 1$ μm, leads to a large tuning of the focal length ($\Delta f \sim 36$ μm change in the front focal length from 781 to 817 μm when the lens separation is changed from 10 to 9 μm, see Supplementary Fig. 1 for the phase profiles and their ray tracing simulations). The membrane and glass lenses are 300 μm in diameter, and have focal lengths of ~120 and ~−130 μm, respectively. The electrostatic actuation is achieved through contacts only to the glass substrate. The capacitor plates are shown in the inset of Fig. 1a. The contacts are configured to make two series capacitors. Each capacitor has one plate on the glass substrate and another one on the membrane, resulting in an attractive force between the membrane and the glass substrate. Figure 1b, c show the first two mechanical resonance modes of the membrane at ~2.6 and ~5.6 kHz, respectively. This limits the operation frequency of the device to ~4 kHz to avoid unwanted excitation of the second resonance.

The metasurfaces are based on high-contrast dielectric transmitarrays[23,27]. These devices consist of arrays of high index dielectric scatterers (nano-posts) with different shapes and sizes. With proper design, the nano-posts enable complete control of phase and polarization on a subwavelength scale[38,40,56]. When only phase control is required, the nano-posts should have a symmetric cross-section (i.e., square, circular, etc.). For fabrication considerations, we choose nano-posts with square-shaped cross-section on a square lattice. Since both the moving and stationary metasurface lenses have high numerical apertures (NA ~0.8), we used a recently developed technique for choosing the metasurface parameters (i.e., amorphous silicon layer thickness, lattice constant, and minimum and maximum post side lengths) to maximize the efficiency of high NA lenses for both transverse electric (TE) and transverse magnetic (TM) polarizations[57]. The method is based on approximating the efficiency of a lens designed with certain metasurface parameters through efficiencies of periodic gratings designed with the same parameters. Using this method and considering the design wavelength of 915 nm, the $\alpha$-Si layer thicknesses were chosen to be 530 and 615 nm for the moving and stationary lenses, respectively. The lattice constant was set to 320 nm in both cases. Figure 1d, e show simulated transmission amplitudes and phases for uniform arrays of nano-posts on the membrane and the glass substrate, respectively. Given a required phase profile, one can find the best nano-post for each site on the metasurface using Fig. 1d, e[27].

**Device fabrication.** A summary of the key fabrication steps for the moving and stationary lenses is schematically depicted in

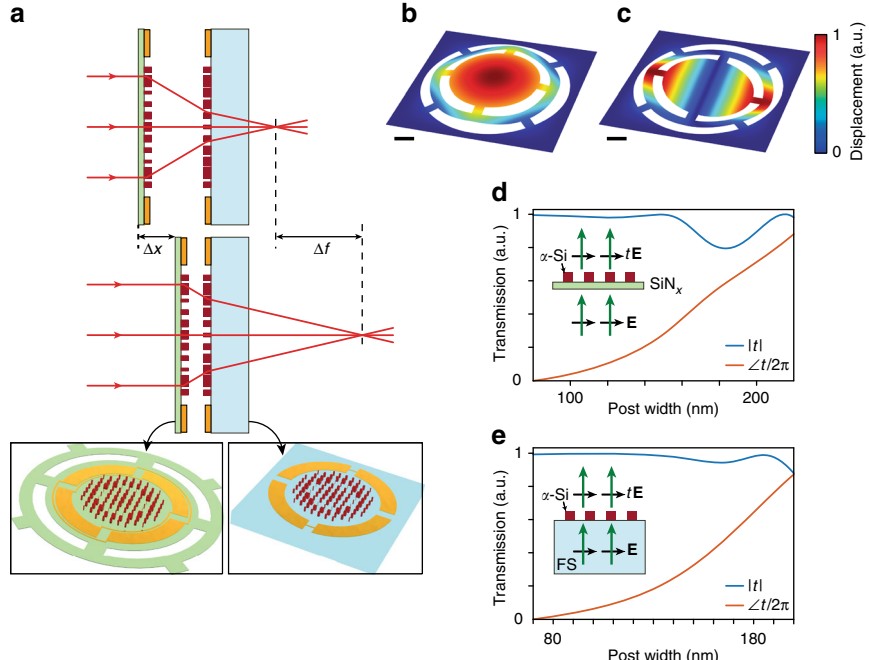

**Fig. 1** Schematic illustration of the tunable doublet and design graphs. **a** Schematic illustration of the proposed tunable lens, comprised of a stationary lens on a substrate, and a moving lens on a membrane. With the correct design, a small change in the distance between the two lenses ($\Delta x \sim 1\,\mu m$) results in a large change in the focal distance ($\Delta f \sim 35\,\mu m$). (Insets: schematics of the moving and stationary lenses showing the electrostatic actuation contacts.) **b** The first and **c** second mechanical resonances of the membrane at frequencies of ~2.6 and ~5.6 kHz, respectively. The scale bars are 100 µm. **d** Simulated transmission amplitude and phase for a uniform array of $\alpha$-Si nano-posts on a ~213-nm-thick $SiN_x$ membrane versus the nano-post width. The nano-posts are 530 nm tall and are placed on the vertices of a square lattice with a lattice constant of 320 nm. **e** Simulated transmission amplitude and phase for a uniform array of $\alpha$-Si nano-posts on a glass substrate versus the nano-post width. The nano-posts are 615 nm tall and are placed on the vertices of a square lattice with a lattice constant of 320 nm. FS: Fused silica

Fig. 2a–f (for more details see the Methods section). The moving metasurface fabrication was started on a silicon wafer with a ~210-nm-thick low-stress $SiN_x$. A 20-nm-thick $SiO_2$ layer followed by a 530-nm-thick $\alpha$-Si layer was deposited on the $SiN_x$ layer. The $SiO_2$ layer acts both as an adhesion promoter between the $SiN_x$ and the $\alpha$-Si layers, and as an etch-stop during the dry etch process to form the metasurface. In the next step, patterns for backside holes were defined and transferred to an alumina layer. This layer was then used as a hard mask to partially etch through the silicon wafer (a ~50-µm-thick layer was left to maintain the mechanical strength of the sample during the next steps). Alignment marks were then etched through the $\alpha$-Si layer for aligning the top and bottom sides. The metasurface lens was then patterned into the $\alpha$-Si layer. Next, the metallic contacts were deposited and patterned. The top side of the device was covered with a protective polymer, and the remaining part of the wafer under the membrane was wet etched. Finally, the membrane was patterned and dry etched to release the metasurface. An optical image of the fabricated metasurface on a membrane is shown in Fig. 2b. Due to the residual stress in the membranes, the beams are slightly bent such that the central part of the lens is about 6 to 8 µm above the surface of the wafer.

The fabrication steps of the stationary metasurface are schematically shown in Fig. 2c. A 615-nm-thick layer of $\alpha$-Si was deposited on a glass substrate. The metasurface pattern was generated and etched through the layer, followed by deposition and patterning of the contacts. An optical image of a completed metasurface on the glass substrate is shown in Fig. 2d. Finally, a 20 µm spacer layer was spin coated and patterned on the glass substrate (to achieve a ~12 µm distance between the lenses), and the two chips were aligned and bonded with an ultraviolet (UV) curable epoxy (Fig. 2e). An optical image of the final device is

shown in Fig. 2f. Figure 2g, h show scanning electron micrographs of the fabricated metasurfaces.

**Experimental doublet characterization results**. Figure 3 summarizes the focusing measurement results under application of a direct-current (DC) voltage. For these measurements, the device was illuminated with a collimated beam from a 915-nm diode laser, and the focal plane intensity patterns were imaged using a custom-built microscope (for details of the measurement setup, see Methods section and Supplementary Fig. 2). The simulated EFL is plotted against the distance between the two lenses in Fig. 3a, along with the measured values for multiple devices with different initial separations between the membrane and the glass substrate. In all measurements, these separations were extracted by comparing the measured focal distances to their simulated values. We should note that devices 2 and 3 are on the same chip, and devices 4–8 are on another chip. This shows the potential of the proposed structure for integrating multiple devices on the same chip with the ability to scan different ranges of focal distances to simultaneously image a larger range of depths. Figure 3b shows the measured front focal length (i.e., the physical distance between the focus and the stationary lens), and the extracted lens separation for device 2 versus the applied DC voltage, indicating that the optical power changes by more than 180 diopters when applying 80 V. The difference between the measured focal distances in a few measurements falls within the measurement error of ~5 µm. The possibility of changing the applied voltage very finely, makes it possible to tune the membrane separation and thus the optical power very finely, in the absence of external vibrations. Intensity distributions measured in the focal plane under application of different DC voltages are shown in Fig. 3c for device 2. As seen in Fig. 3c, d, the measured Airy disk radii are

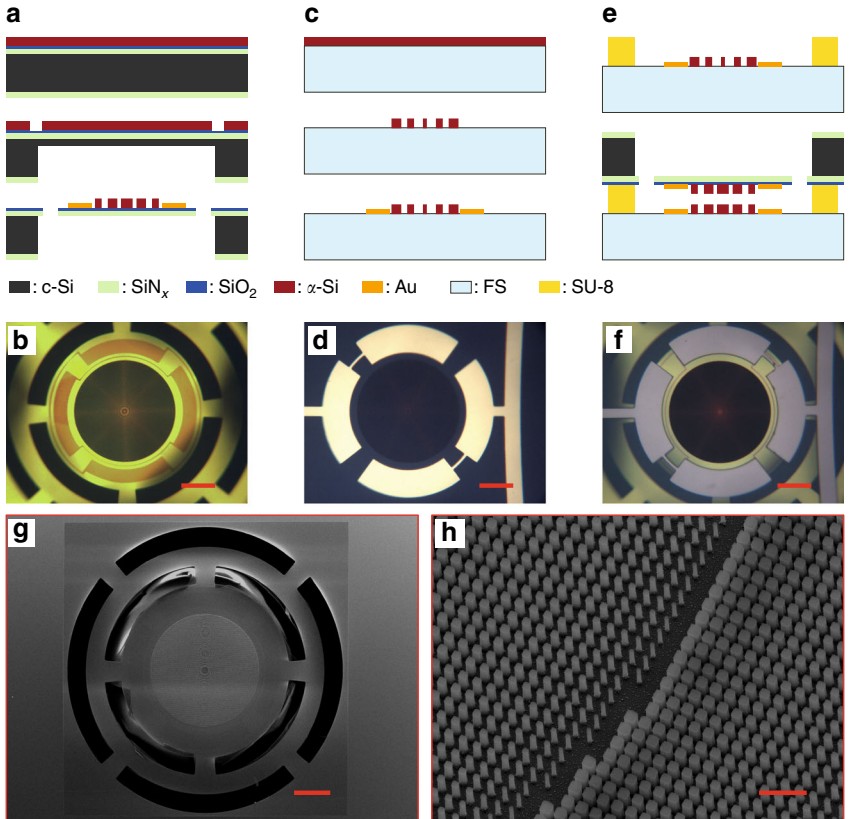

**Fig. 2** Fabrication process summary. **a** Simplified fabrication process of a lens on a membrane: a $SiO_2$ spacer layer and an α-Si layer are deposited on a Si substrate with a pre-deposited $SiN_x$ layer. The backside of the substrate is partially etched, and alignment marks are etched into the α-Si layer. The lens is patterned and etched into the α-Si layer, and gold contacts are evaporated on the membrane. The remaining substrate thickness is etched and the membrane is released. c-Si: crystalline silicon; FS: fused silica. **b** An optical microscope image of a fabricated lens on a membrane. **c** Simplified fabrication process of the lens on the glass substrate: an α-Si layer is deposited on a glass substrate and patterned to form the lens. Gold contacts are evaporated and patterned to from the contacts. **d** An optical microscope image of the fabricated lens on the glass substrate. **e** Schematics of the bonding process: an SU-8 spacer layer is patterned on the glass substrate, the two chips are aligned and bonded. **f** A microscope image of the final device. **g** Scanning electron micrograph of the lens on the membrane, and **h** nano-posts that form the lens. Scale bars are 100 μm in **b**, **d**, **f**, and **g**, and 1 μm in **h**

smaller than 1.1 times their corresponding theoretical values. The observed aberrations are caused by the mechanical deformation of the moving lens resulting from the residual stress in the $SiN_x$ layer. The metasurface lenses are designed for optimal performance when their separation changes from 12 to 6 μm. As a result of this change, the effective focal length should be tuned from 627 to 824 μm. The achieved initial distance between the metasurfaces is slightly different from the design value (~15 μm instead of ~12 μm) because the spacer layer was slightly thicker than intended. Besides, in order to avoid the pull-in instability (which would destroy the device), we stayed away from higher voltage values than 80 V in this sample. In principle, one should be able to decrease the lens separation in device 2 from 15 μm to about 10 μm, and thus increase the front focal distance from 635 to 781 μm (or change the EFL from 560 to 681 μm, tuning optical power by more than 300 diopters, or ~20%).

Figure 3d shows the measured absolute focusing efficiency of the doublet (defined as the power passing through a ~20-μm diameter aperture to the total power hitting the device). The absolute efficiency is between 40 and 45% for all applied voltage values. The high-NA (NA ~0.8) singlets used here are expected to be ~75% efficient[57]. Because the doublet uses two of such lenses, its efficiency is estimated to be ~55%. Taking into account the reflections at the three air-glass interfaces (a glass wafer is used to cap the backside of the membrane to fully isolate it from the environment airflow), we obtain a total efficiency of ~50% which

agrees well with the measured efficiency values. We attribute the slightly lower measured efficiency to fabrication imperfections. It is foreseeable that the efficiency can be significantly improved with better optimization and design processes[35], use of anti-reflection coatings to reduce reflection losses, and optimizing the fabrication process.

The frequency response of the doublet is measured and plotted in Fig. 3e (see Supplementary Note 1 and Supplementary Fig. 2 for details of frequency response measurement). The frequency response transfer function is defined as the membrane displacement at frequency $f$, normalized to its value under the same voltage applied in DC. The black dashed line shows the −3-dB line, showing a ~230 Hz bandwidth for the device. The red and blue dashed lines show second order system fits (i.e., $H(f) = \frac{1}{1 - i\left(b/\sqrt{mk}\right)(f/f_0) - (f/f_0)^2}$, where $f_0$ is the resonance frequency for the first mode, $b$ is a damping factor, and $m$ and $k$ are the oscillator mass and spring constant, respectively), indicating that the fit follows the measurement well for $b = 20\sqrt{mk}$. This corresponds to a highly overdamped system with a damping ratio $(b/2\sqrt{mk})$ of ~10. Under the atmosphere pressure the dominant loss mechanism is the air damping[58]. If the damping is reduced by about 20 times by reducing the air pressure inside the lens packaging (i.e., $b/2\sqrt{mk} \approx 0.5$), then the frequency response will follow the blue dashed line in Fig. 3e, with a 3-dB bandwidth reaching 4 kHz. This would correspond to a quality factor of ~1

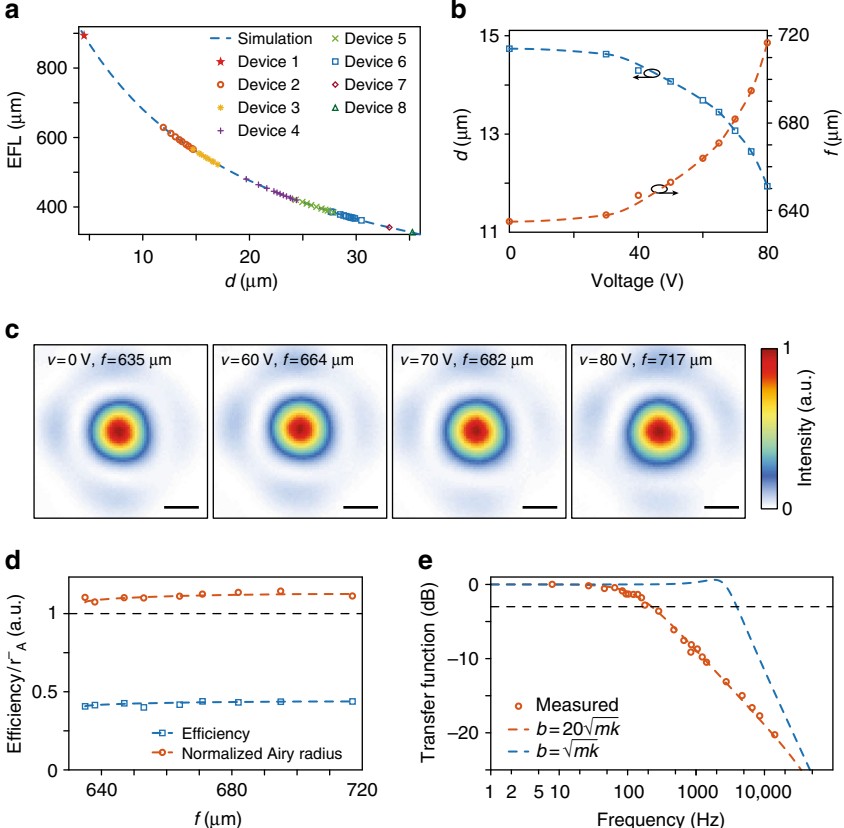

**Fig. 3** Focusing measurement results of the tunable doublet. **a** Simulated EFL versus the distance between lenses, along with measured EFL values for 8 devices under different applied voltages. Different devices have different initial lens separations, resulting in different focal distances under no applied voltage. **b** Measured front focal length versus the applied DC voltage for device 2 of panel **a**. The separation values between the moving and stationary lenses are also plotted. **c** Intensity distributions in the focal plane of the doublet lens at different actuation voltages. The scale bars are 2 μm. **d** Measured Airy radii (normalized to their corresponding diffraction-limited values), $\bar{r}_A$, and measured absolute focusing efficiency of the tunable doublet. **e** Measured frequency response of the system, along with second order transfer functions with two values of the damping factor (b) equal to $20\sqrt{mk}$ and $\sqrt{mk}$

for the mechanical resonator, which should be feasible by reducing air damping. In addition, at such a low quality factor, oscillation and long settling times should not be an issue. Vacuum packaging could be done through bonding the backside glass substrate (the one with no metasurface) and the silicon chip carrying the membrane in a vacuum chamber with controllable pressure.

**Imaging with electrical focusing.** The tunable doublet can be used for imaging with electrically controlled focusing. To demonstrate this, we formed an imaging setup using the doublet and a refractive lens. The setup is schematically shown in Fig. 4a. A transmissive object was placed in front of the imaging system. A 1.8-mm diameter pinhole was placed in front of the aspheric lens to reduce the aperture and increase contrast. The system images the object to a plane ~130 μm outside the stationary lens substrate. Since this image is very small and close to the lens, we used a custom-built microscope (×55 magnification) to re-image it onto the camera. The results are summarized in Fig. 4b. When the object is $p \sim 15$ mm away and no voltage is applied, the image is out of focus. If the applied voltage is increased to 85 V in the same configuration, the image comes to focus. Changing the object distance to $p \sim 9.2$ mm, the voltage should also be changed to 60 V to keep the image in focus. At 0 V, the object should be moved to $p \sim 4$ mm to be in focus, and applying 85 V to the doublet will result in a completely out of focus image in this configuration. As observed here, by moving the membrane only about 4 μm, the overall system EFL changes from 44 to 122 mm, a ratio of about 1:2.8. This is an example of the importance of the

large absolute optical power tunability of the metasurface doublet, especially when it is integrated into a system with a comparably small overall optical power. It also allows for changing the object distance from 4 to 15 mm by electrically controlled refocusing.

**Electrically tunable compact microscope.** To further demonstrate the capabilities of this platform, we use it to design a 1-mm-thick electrically tunable microscope. The structure is schematically shown in Fig. 5a, and is a metasurface triplet composed of a tunable doublet (with an optical design different from the fabricated one), and an additional metasurface lens. The lenses, from left to right are 540, 560, and 400 μm in diameter. They have focal lengths of about −290, 275, and 1470 μm. The glass substrate is 1-mm thick, and the image plane is located 14 mm behind the third lens. The stop aperture is located at the plane of the right-most lens, and has the same diameter of 400 μm. By moving the membrane and changing the separation $d$ from 13 to 5 μm, the object plane distance $D$ changes from 622 to 784 μm. The triplet can be optimized to correct for monochromatic aberrations[50] to keep the focus close to the diffraction limit over a large field of view (see Supplementary Fig. 1 for phase profiles, and Supplementary Table 2 for the corresponding coefficients). Here, we have optimized the phase profiles to keep the focus almost diffraction limited in a ~500 μm diameter field of view (corresponding to a ~40-degree field of view in the $D = 700$ μm case). Spot diagrams of point sources at 0, 125, and 250 μm distances from the optical axis are shown in Fig. 5b, demonstrating a diffraction-limited behavior. The spot diagrams are ray optics

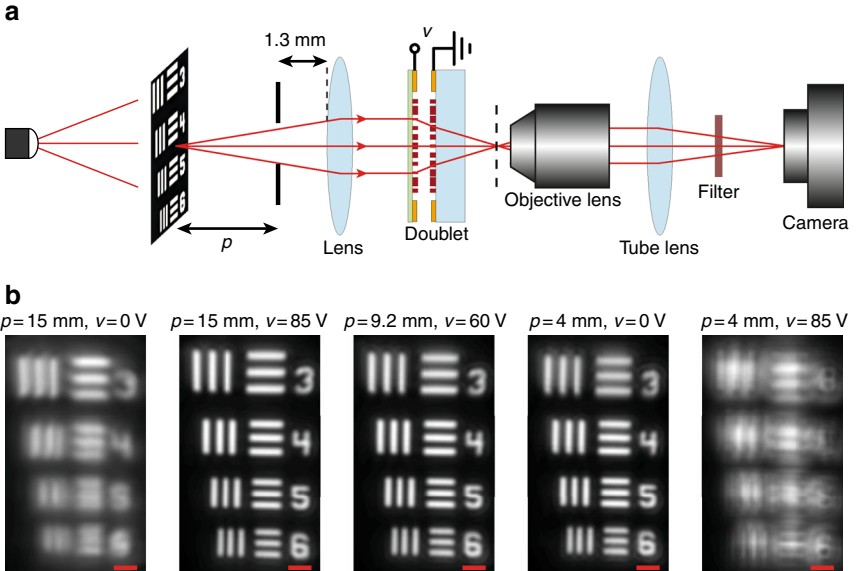

**Fig. 4** Imaging with the tunable doublet. **a** Schematic illustration of the imaging setup using a regular glass lens and the tunable doublet. The image formed by the doublet is magnified and re-imaged using a custom-built microscope with a ×55 magnification onto an image sensor. **b** Imaging results, showing the tuning of the imaging distance of the doublet and glass lens combination with applied voltage. By applying 85 V across the device, the imaging distance $p$ increases from 4 to 15 mm. The scale bars are 10 μm

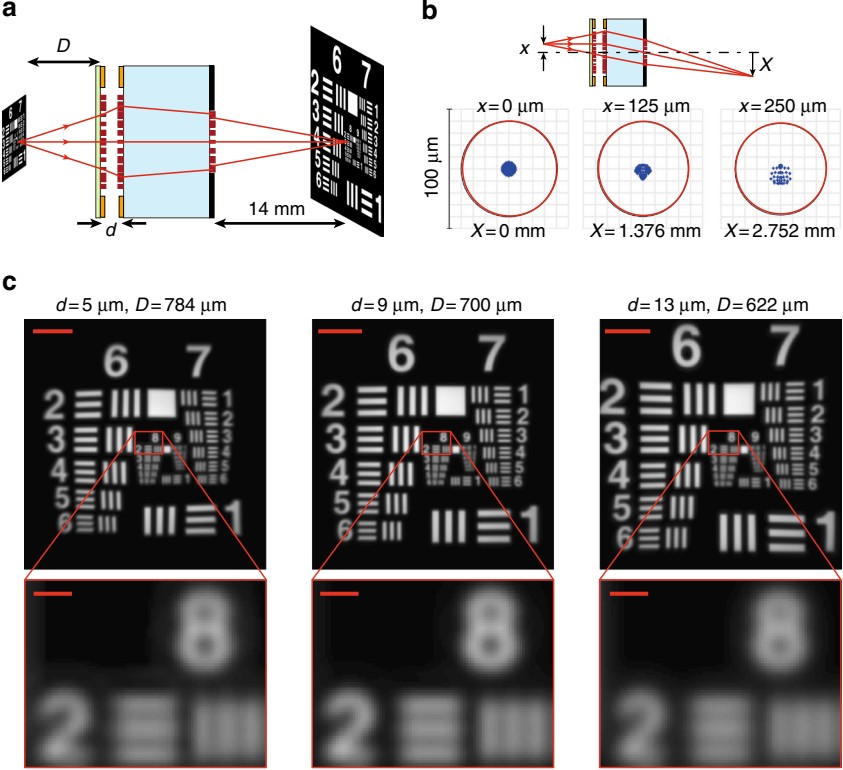

**Fig. 5** Tunable focus metasurface microscope. **a** Schematic illustration of a metasurface triplet operating as a compact electrically tunable microscope. The metasurfaces have diameters of 540, 560, and 400 μm from the left to the right, respectively, and the glass substrate is 1 mm thick. Moving the membrane by about 8 μm moves the object plane more than 160 μm. **b** Ray optics simulation of spot diagrams of the microscope for the case of $d = 9$ μm. The inset shows a schematic of the triplet, the locations of the point source in the object plane and the image plane. The phase profiles of the metasurfaces are designed to keep the focus almost diffraction limited for a 500-μm-diameter field of view when d is changed from 5 to 13 μm. The system has a magnification close to 11 and a numerical aperture of 0.16 when $d = 9$ μm. **c** Image simulation results using the triplet for different values of $d$ and $D$. The scale bars are 50 μm in the zoomed-out images, and 5 μm in the zoomed-in areas

simulation results from a point source at their corresponding distances from the optical axis. The red circles show the diffraction-limited Airy disks for different cases, with ~40-µm radii. Figure 5c shows the image formation simulation results for the system at three different values of $d$ (and $D$). The insets show that the system can resolve the ~3.5 µm line-space in an object. The effective focal length for the whole system is ~1160 µm (for $d$ = 9 µm case), which is significantly larger than the focal lengths of the membrane and first glass lenses, similar to the fabricated doublet. As a result, the object space NA is about 0.16, corresponding to a resolution of ~3.5 µm at the object plane. Considering the 14-mm distance between the image plane and the backside aperture, the image space NA is ~0.014 which results in an Airy radius of about 40 µm in the image plane. As the effective focal length of the system changes with tuning $d$, the total magnification of the system also changes from 11.3 (for $d$ = 5 µm) to 10.3 (for $d$ = 13 µm).

## Discussion

The lenses demonstrated here have small sub-millimeter aperture sizes suitable for applications in ultra-compact optical systems. In principle, the lenses can have centimeter-scale apertures as silicon nitride membranes at these scales have already been demonstrated[59,60]. In addition, the electrostatic forces and the mechanical resonance frequencies can be engineered by appropriate choice of the electrostatic actuation plate areas, membrane thickness, and mechanical beam design.

The high optical power of the elements, and the small aperture of the doublet result in a relatively high sensitivity to the membrane bending, and to misalignment between the two lenses (see Supplementary Fig. 3 for modulation transfer function and Strehl ratio simulation results). We estimate the radius of curvature of the measured membranes to be ~20 mm, using mechanical simulations of the structure and the observed ~6-µm distance between the center of the lens and the surface of the wafer. This would result in a Strehl ratio slightly larger than 0.95. A Strehl ratio of 0.9 (as an acceptability criterion) corresponds to a radius of curvature of ~15 mm. If the membrane curvature is larger than this and known a priori, the lens design can be optimized to include the effects of the bending. In addition, to have a Strehl ratio better than 0.9, the misalignment between the two lenses should be better than 2 µm. On the basis of the symmetric measured focal spots, we estimate the misalignment in the doublets to be smaller than this limit. Considering the high alignment precision achievable with industrial aligners, achieving a 2-µm resolution is not a challenge.

Similar to other diffractive and metasurface optical devices, the lenses demonstrated here suffer from chromatic aberrations[61–63]. The exact "acceptable" operation bandwidth of the lens depends on the effective focal length, the numerical aperture, and a criterion for "acceptability". Using the criterion given in[50] that is based on the focal spot area increasing to twice its value at the center wavelength, and assuming an effective focal length of ~600 µm (corresponding to a numerical aperture of 0.24), the operation bandwidth is given by $\Delta\lambda = 2.27\lambda^2/(f\mathrm{NA}^2) \approx 50$ nm. To make multiwavelength tunable doublets, many of the recently demonstrated approaches for making multiwavelength metasurfaces can be directly applied[56,64–66]. In addition, the recently introduced concept of phase-dispersion control[67–69] can be used to increase the operation bandwidth of the metasurface lenses by correcting the chromatic aberrations over a continuous bandwidth.

Here we introduced a category of MOEMS devices based on combining metasurface optics with the MEMS technology. To showcase the capabilities of the proposed platform, we experimentally demonstrated tunable lenses with over 180 diopters change in the optical power, and measured focusing efficiencies exceeding 40%. In principle, the optical power tunability could be increased to above 300 diopters for the presented design. We demonstrated how such tunable lenses can be used in optical systems to provide high-speed electrical focusing and scanning of the imaging distance. The potentials of the introduced technology go well beyond what we have demonstrated here, and the devices can be designed to enable compact fast-scanning endoscopes, fiber-tip-mounted confocal microscopes, etc. In principle, metasurfaces can replace many of the refractive and diffractive microoptical elements used in conventional MOEMS devices to make them more compact, increase their operation speed, and enhance their capabilities.

## Methods

**Simulation**. The optimized phase profiles (for both the fabricated doublet and the triplet shown in Fig. 5) were obtained using Zemax OpticStudio. The phase profiles are defined as even-order polynomials of the radial coordinate $r$ according to $\phi(r) = \sum_{2n} a_{2n}(r/R_0)^{2n}$, where $R_0$ is a normalization radius and $a_{2n}$ are the coefficients (see Supplementary Fig. 1 and Supplementary Table 1 for the phase profiles and the optimized coefficients). This was done through simultaneously minimizing the root mean square radius of the focal spot for several configurations (i.e., different lens separations, and, in the case of the triplet, the lateral source position). The image formation simulations (for Fig. 5) were done using the extended scene simulations of Zemax OpticStudio and took into account the aberrations and limitations resulting from diffraction.

Mechanical simulation of the MEMS structure was performed in COMSOL Multiphysics to find the resonances of the structure. The metallic contacts and the $\alpha$-Si metasurface were treated as additional masses on the membrane. The Young modulus of SiN$_x$ was assumed to be 250 GPa and its Poisson ratio was set to be 0.23. The following densities were used for different materials: 3100 kg m$^{-3}$ for SiN$_x$, 2320 kg m$^{-3}$ for $\alpha$-Si, and 19,300 kg m$^{-3}$ for gold. To account for the fact that the whole metasurface volume is not filled with $\alpha$-Si, an average fill factor of 0.5 was used.

Transmission amplitudes and phases of the metasurface structures on both fused silica and silicon nitride were computed through rigorous coupled-wave analysis[70]. The transmission values were calculated by illuminating a uniform array of nano-posts with a normally incident plane wave at 915 nm wavelength and finding the amplitude and phase of the transmitted zeroth-order wave right above the nano-posts. The subwavelength lattice ensures that this single number is adequate to describe the optical behavior of a uniform array. The following refractive indices were used in the simulations: 3.5596 for $\alpha$-Si, 2.1 for SiN$_x$, 1.4515 for fused silica. The lattice constant was 320 nm in both cases, and the $\alpha$-Si thickness was 530 nm for the moving, and 615 nm for the stationary lens.

**Device fabrication**. Fabrication of the stationary lenses was started by depositing a 615-nm-thick layer of $\alpha$-Si on a 500-µm-thick fused silica substrate through a plasma enhanced chemical vapor deposition (PECVD) process. The metasurface pattern was written on a ~300-nm-thick layer of ZEP520A positive electron resist with a Vistec EBPG5000+ electron beam lithography system. After development of the resist, a 70-nm-thick alumina layer was evaporated on the sample that was used as a hard mask. The pattern was then transformed to the $\alpha$-Si layer via a dry etch process. The metallic contacts' pattern was defined using photolithography on AZ 5214 E photoresist which was used as a negative resist. A ~10-nm-thick layer of Cr, followed by a ~100-nm-thick Au layer was evaporated onto the sample, and a lift-off process transferred the photoresist pattern to the metal layer. Finally, a ~20-µm-thick layer of SU-8 2015 was spin coated on the sample and patterned to function as a spacer.

The moving lens fabrication started with a silicon wafer with ~450-nm-thick low-stress low-pressure chemical vapor SiN$_x$ deposited on both sides. The device side was etched down to about 213 nm with a dry etch process. A ~20-nm-thick SiO$_2$ layer, followed by a ~530-nm-thick $\alpha$-Si layer was deposited on the sample with a PECVD process. Through hole patterns were defined on the backside of the sample using the AZ 5214 E photoresist, and a lift-off process was performed to transfer the pattern to a ~200-nm-thick alumina layer that was used as a hard mask. The holes were partially etched through the wafer with a Bosch process (leaving a ~50-µm-thick silicon layer to provide mechanical support for the membrane in the following steps). Alignment marks (for aligning the lenses to the backside holes) were patterned and etched into the $\alpha$-Si layer using a backsidealigned photolithography process. A process similar to the one used for the stationary lenses was performed to fabricate the metasurfaces and the metallic contacts. The top side (with the metasurfaces and contacts) was then covered with a protective polymer coating (ProTEK PSB, Brewer Science) layer, and the remaining ~50-µm-thick silicon layer was etched in a 3:1 water-diluted potassium hydroxide solution at 80 °C. The membrane pattern was defined on the sample using photolithography with AZ nLOf 2020 photoresist, and was etched through the SiN$_x$ membrane to release the membrane. The photoresist was then removed in

an oxygen plasma. A fused silica piece was bonded to the backside of the membrane sample using a UV-curable epoxy (NOA 89, Norland Products) to isolate the membranes from ambient airflow. At the end, the moving and stationary samples were aligned and bonded using an MA6/BA6 aligner (Suss MicroTec). A UV-curable epoxy was used to bond the two samples. Using this technique, an alignment precision of a few microns is feasible.

**Measurement procedure.** The doublet characterization setup is schematically shown in Supplementary Fig. 2a. A collimated beam from a fiber coupled 915-nm diode laser connected to a fiber collimation package (F240FC-B, Thorlabs) was used to illuminate the doublet from the membrane side. A custom-built microscope consisting of a ×50 objective (Olympus LCPlanFL N, NA = 0.7) and a tube lens with a 20-cm focal length was used to image the focal plane of the doublet to a charge-coupled device camera (CoolSNAP K4, Photometrics). An air coplanar probe (ACP40 GSG 500, Cascade Microtech) was used to apply a voltage to the doublet. For measuring the frequency response, square pulses with different base frequencies were applied to the probe (CFG250 function generator, Tektronix). The change in the optical power passing through a 50-μm pinhole in the image plane (equivalent to a ~1-μm pinhole in the focal plane) was then measured with a fast detector (PDA36A, Thorlabs) connected to an oscilloscope. The frequency response was then extracted through Fourier transforming the input voltage and the resulting change in the output power (see Supplementary Note 1 for more details).

The efficiency was calculated through measuring the power passing through a ~1-mm iris in the image plane (corresponding to a ~20-μm pinhole in the focal plane) and dividing it by the total power before the doublet. To make sure that the total beam was incident on the doublet, the beam was partially focused by a lens with a 10-cm focal length. The distance between the lens and the doublet was adjusted such that the beam had a ~100-μm full-width at half-maximum (FWHM) at the place of the doublet (i.e., one third of the doublet diameter). This way, more than 99% of the power is expected to hit the doublet area.

The imaging experiments in Fig. 4 were also performed using a similar setup. For imaging, a 910-nm LED (LED910E, Thorlabs) was used as an illumination. To reduce the effects of chromatic dispersion, a bandpass filter (FB910-10, 910-nm center wavelength, 10-nm FWHM) was placed in front of the camera. A negative 1951 USAF Resolution target (R1DS1N, Thorlabs) was used as an imaging object. A 4-f system consisting the doublet and a glass lens with focal length of 8 mm (ACL12708U-B, Thorlabs) was used to form images of the resolution target at different distances. To reduce the aperture size and increase contrast, a 1.8-mm-diameter aperture (AP1.5, Thorlabs) was placed at a ~1.3-mm distance in front of the refractive lens. The distance between the backside of the refractive lens and the doublet was ~4.5 mm. The resulting image was magnified and re-imaged onto the camera with the same microscope used for the focal spot characterization.

**Data availability**. The data that support the findings of this study are available from the corresponding author upon request.

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

## Acknowledgements

This work was supported by National Science Foundation award 1512266 and Samsung Electronics. A.A. and Y.H. were also supported by DARPA, and S.M.K. was supported as part of the Department of Energy (DOE) "Light-Material Interactions in Energy Conversion", Energy Frontier Research Center under grant no. DE-SC0001293. The device nanofabrication was performed at the Kavli Nanoscience Institute at Caltech.

## Author contributions

A.A., A.F., and E.A. conceived the experiment. E.A., A.A., S.M.K., and Y.H. fabricated the samples. E.A., A.A., S.M.K., and M.F. performed the simulations, measurements, and analyzed the data. E.A., A.A., and A.F. co-wrote the manuscript. All authors discussed the results and commented on the manuscript.

## Additional information

**Competing interests:** The authors declare no competing financial interests.

