## [Peer Review File · Nature Communications]

Reviewers' comments:

Reviewer #1 (Remarks to the Author):

This paper presents discussion and demonstration of a varifocal lens system utilizing MEMS technology to control the longitudinal distance between two thin lens profiles implemented as dielectric metasurfaces. Control of the distance between the metasurfaces controls the overall optical power of the system. Very basic discussion of the operation of the system is presented, along with more detailed discussion of fabrication and testing/characterization. The paper is well written with good figures and reasonable detail in the supplementary documents. This work is likely to be of interest to others in the field of metasurfaces, MEMS, and imaging systems.

The intended novelty of the work is not explicitly stated, but is inferred to be the demonstration of mechanically tunable optical systems based on metasurfaces. I note that the authors have already demonstrated the use of metasurface lens doublets for imaging systems (Ref. 52), and other researchers have demonstrated the uses of MEMS and metasurfaces to vary optical functions (Refs. 54-55). With this in mind, the work in this paper is of high quality, but not necessarily novel in the context of this prior work.

I have several additional comments and questions.

1.The authors discuss that a large change in optical power can be achieved by a very small movement of the membrane (e.g., 60 diopters from a 1 μm movement). However, they do not discuss the negative of this capability....what level of movement control is required and achievable for finer control of the system power?

2.The discussion of the optical power tunability is inconsistent. On line 138 they discuss demonstrated changes in optical power of 180 diopters, and on line 151 they discuss (but do not demonstrate a scenario for a 300 diopter change. However, on line 207 the authors state 'we demonstrated tunable lenses with over 300 diopter change in the optical power.' This claim is misleading.

3.The authors discuss testing of the components with lasers in the 900 nm range. However, I cannot find any discussion as to the wavelength value or range used for the design of the components themselves. This is very relevant for the design of the sub-wavelength features that make up the metasurface. The authors mention (line 199) that the lenses suffer from chromatic aberrations, but do not quantify the statement in any way.

4.The aperture sizes discussed in the work are extremely small, nominally a consequence of the fabrication and actuation approaches used. Some discussion as to the scalability limits on aperture size is warranted so that the broader applicability of this technology can be assessed.

5.In Figure 3, experimental measurement data are presented, but there is no mention as to how many measurements were performed or to the statistics of the measurements.

Reviewer #2 (Remarks to the Author):

E. Arbabi et al., are proposing a varifocal optical system based on MEMS-activated metasurfaces. The authors have demonstrated that by combining two metalenses with large and opposite-sign optical powers, it is possible to achieve large tuning of the optical power even with very small movements of only one of the two metasurfaces. Positioning of the metalens is performed via

electrostatic forces applied on a membrane. Several devices with different separation distances between the two lenses have been fabricated and their focusing properties carefully characterized. These tunable metalens systems have been used for imaging.

The technique is greatly inspired by works performed more than ten years ago by several groups working on MEMS devices. In terms of technology, the results presented in this paper are really interesting and deserve publication but not in a general audience journal as Nature Communications which usually reports novel phenomenon.

I recommend resubmitting the work to specialized journal like Journal of Microelectromechanical Systems or Journal of Micro/Nanolithography, MEMS, and MOEMS.

Reviewer #3 (Remarks to the Author):

This paper describes the integration of flat meta-surface lenses with MEMS actuators to make the miniature flat optics tunable. This is perhaps the first report of MEMS tunable meta-surface flat lenses, albeit not exactly to make a single lens tunable but rather make the effective focal length of a pair of meta-surface lenses tunable through MEMS induced gap changes. The integration of ultra-thin flat meta-surface lenses with MEMS for dynamic tunability has a number of advantages including fast tuning speed. The concept of this tuning method is not new but results are quite impressive, especially when combined with a third meta-surface lens to demonstrate a chip scale microscope only 1 mm thick and yet is able to have dynamic tunability to focus onto different layers. This chip scale microscope would have impact on biomedical and life sciences.

However, the following major issues are also raised:

1) The emphasis of extremely large change of optical power (hundreds of diopters) over 1 micron movement is quite misleading. Here, the authors are actually utilizing the scaling of a conventional two-lens combination down to microscale using nanofabrication and meta-surface designs. Quite obviously, this WILL lead to large change of diopters caused by small axial movement, because now the focal lengths are shrunk down to micron scale and optical powers increased to thousands of diopters. At this high value of optical power, it is actually relatively easy to induce a variation of hundreds of dioptres. However, when looking at the performances of a scaled-down optical system, this large change of diopters does not actually help the miniaturized system to achieve extra tunability. For example, the authors reported "change the EFL from 560 μm to 681 μm , tuning optical power by more than 300 diopters" on page 6 line 150, however, when we examine the variation ratios of focal lengths, optical magnifications, etc., they are actually around 1.2X. I thus suggest the authors rephrase these statements and if possible add some discussions.

2) In the fabrication process of the moveable lens, what is the purpose of using the 20 nm SiO₂ layer? The authors stated that there are residual stresses in their Si₃N₄ membranes and the central area is bent out about 5 to 6 microns. Please provide the surface profiler results of the movable lens in released state, and provide the estimated radius of curvature, and discuss the effects/impacts of this undesirable curvature on the image quality.

3) The authors used 530 nm and 615 nm alpha-Si layer thicknesses respectively for movable and stationary lenses. Why they are chosen differently and are there any design considerations here?

4) The movable lens chip and stationary lens chip are aligned and bonded using UV-epoxy. What is the alignment accuracy here? When scaling down a thin lens pair to micrometre scale, the required axial movement to achieve a desirable EFL variation ratio is drastically reduced and so is the lens aperture. The reduction in lens aperture results in an increased performance sensitivity to misalignment errors. It is expected that slight misalignment may induce a significant reduction in imaging quality. It is suggested that the authors analyse and discuss such effects.

5) On page 6 line 140, the authors measured "the focusing is close to the diffraction limit (with the

measured Airy disk radius smaller than 1.1 times the theoretical value). The slight aberrations are caused by the mechanical deformation of the moving lens resulting from the residual stress in the SiNx layer." I do not entirely agree here. When scaling down an optical system, the geometric aberration scales down significantly as well, however diffraction-limited Airy radius does not change. The authors measured 1.1 times bigger spot than the diffraction limit, which indicates the aberrations of the lens are not slight indeed.

6) The efficiency of the reported doublet is not high even from theoretical estimations. Is there any way to enhance this efficiency through design? For conventional lenses, efficiency can be enhanced through anti-reflection coatings, how about such meta-surface flat lenses? Please discuss in the paper.

7) On page 7 line 168, the authors stated that if the damping is reduced 20 times, the response of the device can reach 4kHz. Please briefly discuss how to control damping accurately. As far as I know, low pressure or vacuum packaging of MEMS devices usually results in oscillations and increased settling time. In addition, for optical imaging applications such low pressure packaging seems not possible because the focal length is short here and object plane is quite close to the lens surface, which makes vacuum packing challenging.

8) The description of the triplet microscope is unclear. Where is the system aperture located? And how large the aperture is? I guess it is located at the last surface of the system (the same plane as the third meta-surface lens), but this must be clearly stated. In addition, why such a triplet system has a low numerical aperture of only 0.16, while the previous descriptions of the doublet lenses clearly indicate the numerical aperture around 0.8 for the doublets? I guess this is because of the optical design? Please discuss. And, why the diffraction limited Airy disks have about 40-micron radii, this is quite large even for 910 nm wavelength light?

Overall, this paper can be recommended to be accepted for publication only when all the above issues are cleared.

Here we include our responses to the reviewers' comments and denote the changes made in the manuscript. For the sake of readability, our responses are printed in blue font and the changes to the manuscript are presented in green.

Reviewers' comments:

Reviewer #1 (Remarks to the Author):

This paper presents discussion and demonstration of a varifocal lens system utilizing MEMS technology to control the longitudinal distance between two thin lens profiles implemented as dielectric metasurfaces. Control of the distance between the metasurfaces controls the overall optical power of the system. Very basic discussion of the operation of the system is presented, along with more detailed discussion of fabrication and testing/characterization. The paper is well written with good figures and reasonable detail in the supplementary documents. This work is likely to be of interest to others in the field of metasurfaces, MEMS, and imaging systems.

Our response: We thank the reviewer for the careful consideration of the manuscript, for the constructive feedback, and for finding it to be of interest to a broad readership.

The intended novelty of the work is not explicitly stated, but is inferred to be the demonstration of mechanically tunable optical systems based on metasurfaces. I note that the authors have already demonstrated the use of metasurface lens doublets for imaging systems (Ref. 52), and other researchers have demonstrated the uses of MEMS and metasurfaces to vary optical functions (Refs. 54-55). With this in mind, the work in this paper is of high quality, but not necessarily novel in the context of this prior work.

Our response: The novelty of the paper is two folded: first, it demonstrates a fast, highly tunable lens system based on the integration of metasurface lenses with MEMS actuators. It is worth noting in this regard that characteristics of dielectric metasurfaces (i.e., the thin form factor and light weight, in addition to the ability to control the phase with high accuracy and high gradients, while keeping the efficiency high) are key enablers in this structure. The thin form factor and high phase gradients allow for the large tunability, and the light weight enables fast movement. In addition, multiple of these tunable lenses with different designs can be integrated on the same chip and fabricated in the same process. Second, the manuscript introduces a new MOEMS platform based on the integration of metasurfaces and the MEMS technology. We anticipate the applications of this platform to go far beyond what we have demonstrated here. We also note here that the previous demonstrations of MEMS-tunable "metasurfaces" (including Refs. 60-62, cited in the paper for the sake of completeness), have been limited to the (rather simple) case of tuning the resonance wavelength of Fabry-Perot cavities, or propagation length of light by moving a uniform high contrast grating mirror. We have edited the introduction part to reflect this:

"To date, integration of metasurfaces and MEMS devices has been limited to moving uniform high-contrast grating mirrors to tune the resonance wavelength of Fabry-Perot cavities [60, 61], or change roundtrip propagation length of light to form spatial light modulators [62]."

I have several additional comments and questions.

1.The authors discuss that a large change in optical power can be achieved by a very small movement of the membrane (e.g., 60 diopters from a 1 μm movement). However, they do not discuss the negative of this capability....what level of movement control is required and achievable for finer control of the system power?

Our response: As seen in Fig. 3b, the membrane moves about 4 μm upon changing the applied voltage by about 40V (from 40 V to 80 V). In the absence of external vibrations, and far from the pull-in voltage (which is above 85 V for the lens measured in Fig. 3b) the positioning of the lens is very accurate (with 0.1 V corresponding to about 0.01 μm movement), and repeatable upon multiple trials. Since the input voltage can be very finely tuned, the smallest tunability is most likely limited by the finite depth of focus than by the movement of the membrane. In addition, the proportionality coefficient of the electrostatic force to the voltage can be controlled by changing the capacitor plane areas. To clarify this, we have added the following to page 6:

“The possibility of changing the applied voltage very finely, makes it possible to tune the membrane separation and thus the optical power very finely, in the absence of external vibrations.”

2.The discussion of the optical power tunability is inconsistent. On line 138 they discuss demonstrated changes in optical power of 180 diopters, and on line 151 they discuss (but do not demonstrate a scenario for a 300 diopter change. However, on line 207 the authors state ‘we demonstrated tunable lenses with over 300 diopter change in the optical power.’ This claim is misleading.

Our response: We have corrected the discussion section in the following way to more exactly reflect the findings of the manuscript:

“To showcase the capabilities of the proposed platform, we experimentally demonstrated tunable lenses with over 180 diopters change in the optical power, and measured focusing efficiencies exceeding 40%. In principle, the optical power tunability could be increased to above 300 diopters for the presented design.”

3.The authors discuss testing of the components with lasers in the 900 nm range. However, I cannot find any discussion as to the wavelength value or range used for the design of the components themselves. This is very relevant for the design of the sub-wavelength features that make up the metasurface.

Our response: The lenses are designed for operation at 915 nm. The results of Fig. 3 are measured with a laser diode at ~ 915 nm, and the imaging experiments of Fig. 4 are done with an LED with a 910-nm center wavelength, filtered by a 10-nm bandwidth filter centered at the same wavelength. The measurement details are in the methods section, and we have added the following to the beginning of page 5 to clarify the design wavelength:

“Using this method and considering the design wavelength of 915 nm, the α -Si layer thicknesses were chosen to be ...”.

The authors mention (line 199) that the lenses suffer from chromatic aberrations, but do not quantify the statement in any way.

Our response: We have extensively studied the chromatic dispersion properties of metasurface lenses in previous works (Refs. 58, 72, and 78), and it is not in the scope of the present manuscript to study these

effects. Nevertheless, we have added the following to the discussion to quantify the “acceptable operation bandwidth” of the lens based on a criterion presented in the Supplementary Information of Ref. 58:

“The exact “acceptable” operation bandwidth of the lens depends on the effective focal length, the numerical aperture, and a criterion for “acceptability”. Using the criterion given in [58] that is based on the focal spot area increasing to twice its value at the center wavelength, and assuming an effective focal length of $\sim 600 \mu\text{m}$ (corresponding to a numerical aperture of 0.24), the operation bandwidth is given by $\Delta\lambda = 2.27 \lambda^2 / (f \text{NA}^2) \approx 50 \text{ nm}$.”

4. The aperture sizes discussed in the work are extremely small, nominally a consequence of the fabrication and actuation approaches used. Some discussion as to the scalability limits on aperture size is warranted so that the broader applicability of this technology can be assessed.

Our response: The main application areas that we anticipate for this technology are in ultra-compact optical systems which justify the small submillimeter aperture sizes. Nevertheless, from a fabrication point of view, centimeter scale silicon nitride membranes with thicknesses in the few hundreds of nanometers have already been demonstrated (see “*AIP Advances 6, 065004 (2016)*” and “*arXiv:1707.08128v1*”, for example). In addition, the electrostatic forces scale with the area, therefore with correct scaling of the membrane thickness and widths of the mechanical beams, similar voltages can be used for actuation of the membrane. Finally, the mechanical resonance frequencies decrease with increasing the area. However, using a thicker membrane, and designing beams with a thin layer of silicon remaining on the bottom of the beams can increase these frequencies. To address these points, we have added the following paragraph to the discussion section of the manuscript:

“The lenses demonstrated here have small sub-millimeter aperture sizes suitable for applications in ultra-compact optical systems. In principle, the lenses can have centimeter-scale apertures as silicon nitride membranes at these scales have already been demonstrated [67, 68]. In addition, the electrostatic forces and the mechanical resonance frequencies can be engineered by appropriate choice of the electrostatic actuation plate areas, membrane thickness, and mechanical beam design.”

5. In Figure 3, experimental measurement data are presented, but there is no mention as to how many measurements were performed or to the statistics of the measurements.

Our response: There are multiple types of experimental data presented in Fig. 3. More information about the frequency response measurements is given in the Supplementary section S1 and Fig. S2. Frequency response measurements were performed at four different intervals over the span of a few minutes, and the data was averaged to obtain the frequency response. For the focal distance data in Figs. 3a-3c, three measurements were performed and the difference between different measurements falls within the measurement error ($\sim 5 \mu\text{m}$), limited by the precision of the objective positioning and the accuracy of measuring the maximum intensity point. No measureable difference in the Airy disk radii were observed in the three measurements. We have added the following to page 6 to clarify this: “The difference between the measured focal distances in a few measurements falls within the measurement error of $\sim 5 \mu\text{m}$.”

Reviewer #2 (Remarks to the Author):

E. Arbabi et al., are proposing a varifocal optical system based on MEMS-activated metasurfaces. The authors have demonstrated that by combining two metalenses with large and opposite-sign optical powers, it is possible to achieve large tuning of the optical power even with very small movements of only one of the two metasurfaces. Positioning of the metalens is performed via electrostatic forces applied on a membrane. Several devices with different separation distances between the two lenses have been fabricated and their focusing properties carefully characterized. These tunable metalens systems have been used for imaging.

The technique is greatly inspired by works performed more than ten years ago by several groups working on MEMS devices. In terms of technology, the results presented in this paper are really interesting and deserve publication but not in a general audience journal as Nature Communications which usually reports novel phenomenon.

I recommend resubmitting the work to specialized journal like Journal of Microelectromechanical Systems or Journal of Micro/Nanolithography, MEMS, and MOEMS.

Our response: We thank the reviewer for carefully reading the manuscript and we are happy that the manuscript was found technologically interesting. As discussed in the introduction part of the manuscript, we acknowledge that the current manuscript and the works mentioned by the reviewer (including Refs. [8-10, 63]) have conceptual analogies to some degree. In essence, they are all based on miniaturization of varifocal lenses by combining micro-optics with microelectromechanical systems. However, as the manuscript findings show, the use of dielectric metasurfaces enables operation regimes and principles inaccessible with conventional micro-optics. For example, as shown here, the thin form factor and the ability to control the phase with very high precision and large gradients (i.e. high numerical apertures) allows for very high tunabilities. This is in contrast to previous works, where the change in the focal distance is generally of the same order as the mechanical movement of the lens. In addition, the control over the design and fabrication of the conventional micro-optical elements is extremely limited compared to metasurfaces that enable full and arbitrary control of polarization and phase with very high precision. Generally, we believe this work will facilitate and encourage the advent of a new interdisciplinary field based on integrating metasurfaces into the established MEMS technology. We anticipate that given the freedom of design and implementation brought about by metasurfaces, this field will have applications far exceeding what is demonstrated in this manuscript or with conventional micro-optics. As a result, we respectfully agree with the other two reviewers that find the work of interest to a broad readership outside the MEMS community.

Reviewer #3 (Remarks to the Author):

This paper describes the integration of flat meta-surface lenses with MEMS actuators to make the miniature flat optics tunable. This is perhaps the first report of MEMS tunable meta-surface flat lenses, albeit not exactly to make a single lens tunable but rather make the effective focal length of a pair of meta-surface lenses tunable through MEMS induced gap changes. The integration of ultra-thin flat meta-surface lenses with MEMS for dynamic tunability has a number of advantages including fast tuning speed. The concept of this tuning method is not new but results are quite impressive, especially when combined with a third meta-surface lens to demonstrate a chip scale microscope only 1 mm thick and

yet is able to have dynamic tunability to focus onto different layers. This chip scale microscope would have impact on biomedical and life sciences.

Our response: We are grateful to the reviewer for carefully considering the manuscript and for providing constructing feedback. We are glad that the reviewer found the manuscript results impressive.

However, the following major issues are also raised:

1) The emphasis of extremely large change of optical power (hundreds of diopters) over 1 micron movement is quite misleading. Here, the authors are actually utilizing the scaling of a conventional two-lens combination down to microscale using nanofabrication and meta-surface designs. Quite obviously, this WILL lead to large change of diopters caused by small axial movement, because now the focal lengths are shrunk down to micron scale and optical powers increased to thousands of diopters. At this high value of optical power, it is actually relatively easy to induce a variation of hundreds of dioptres. However, when looking at the performances of a scaled-down optical system, this large change of diopters does not actually help the miniaturized system to achieve extra tunability. For example, the authors reported “change the EFL from 560 μm to 681 μm , tuning optical power by more than 300 diopters” on page 6 line 150, however, when we examine the variation ratios of focal lengths, optical magnifications, etc., they are actually around 1.2X. I thus suggest the authors rephrase these statements and if possible add some discussions.

Our response: The reviewer correctly mentions that the achieved tunabilities are the result of the high optical powers of the elements and the very small separation between the lenses (as discussed in introduction, the small separation between the lenses is also necessary to achieve large tunability). While this seems theoretically trivial (simply using paraxial thin lens formulations), there are practical and physical problems rendering it very challenging if one were to use regular refractive micro-optics instead of metasurfaces. For instance, polymer lenses (as usually used in MOEMS) with the same aperture and focal distances of the doublet presented here would each be about 150 μm thick. It is clear that putting such “large elements” at a 10- μm distance would be extremely challenging, if at all possible. This, in addition to the lack of the ability to finely control the surface profile of such refractive microlenses, is responsible for the limited tunabilities previously reported for MOEMS systems. In contrast, the dielectric metasurface platform we have used here solves both of these problems by allowing very fine and precise control of the phase to make a lens with a very high power and sub-micron thickness.

In addition, we believe the absolute change in the optical power does actually bear some importance, especially when the doublet is integrated into a system with an overall power significantly smaller than the doublet. For instance, the system in Fig. 4 shows the object distance can be moved from 4 mm to 15 mm (the overall system EFL changes from ~ 122 mm to ~ 44 mm, upon moving the membrane only 4 μm). Nevertheless, to address the reviewer’s comment and emphasize that the normalized optical power changes are still on the order of a few percent for a 1- μm movement of the membrane, we have rephrased the statements mentioned by the reviewer:

Abstract: “Here, we propose and demonstrate MEMS-based tunable metasurface doublets with more than 60 diopters (about 4%) change in the optical power upon a 1- μm movement of a membrane with one of the metasurface elements.”

Introduction: “The large and opposite-sign optical powers of the two elements, as well as their very close proximity, make it possible to achieve large tuning of the optical power (~ 60 diopters, corresponding to about 4%) with small movements of one element (~ 1 micron).”

Page 6: “In principle, one should be able to decrease the lens separation in device 2 from 15 μm to about 10 μm , and thus increase the front focal distance from 635 μm to 781 μm (or change the EFL from 560 μm to 681 μm , tuning optical power by more than 300 diopters, or $\sim 20\%$).”

We have also added the following to page 8 to emphasize that in an optical system with an overall power that is small compared to that of the doublet, the resulting tunabilities can in fact be significant: “As observed here, by moving the membrane only about 4 μm , the object distance can be moved from 4 mm to 15 mm (corresponding to the overall EFL of the system changing from 44 mm to 122 mm, a ratio of about 1:2.8). This is an example of the importance of the large absolute optical power tunability of the metasurface doublet, especially when it is integrated into a system with a comparably small overall optical power.”

2) In the fabrication process of the moveable lens, what is the purpose of using the 20 nm SiO_2 layer?

Our response: The SiO_2 layer acts both as an adhesion promoter between the SiN_x layer and the a-Si layer, and as an etch-stop layer when etching the a-Si layer to form the metasurface.

“The SiO_2 layer acts both as an adhesion promoter between the SiN_x and the $\alpha\text{-Si}$ layers, and as an etch-stop during the dry-etch process to form the metasurface.”

The authors stated that there are residual stresses in their Si_3N_4 membranes and the central area is bent out about 5 to 6 microns. Please provide the surface profiler results of the movable lens in released state, and provide the estimated radius of curvature, and discuss the effects/impacts of this undesirable curvature on the image quality.

Our response: Unfortunately, it is not possible for us to measure the surface profile of the membrane for two reasons: first, the samples are bonded, and separating the membrane and glass samples will most likely damage the membranes. Second, we don't have access to a non-contact profilometer, and a contact profilometer would damage the membrane (which is only ~ 200 nm thick), or change its surface profile under pressure. Therefore, we have estimated the radius of curvature through simple calculations and mechanical simulations of the structure to be about 20 mm. We note here that the ~ 6 - μm distance is between the center of the lens and the surface of the wafer (which includes the bending of the mechanical beams). The difference in the height of the lens at the center and the edges is beyond the axial resolution of the measurement microscope as they are both in focus simultaneously (from calculations and simulations, we estimate this distance to be ~ 0.5 μm , corresponding to the ~ 20 -mm radius of curvature). To discuss the effects of curvature, we have added Fig. S3a to the Supplementary information, where simulated results of the modulation transfer function and Strehl ratio are plotted for a few values of radius of curvature. We have also added the following to the discussion section to address the effect:

“The high optical power of the elements, and the small aperture of the doublet result in a relatively high sensitivity to bending of the membrane, and to misalignment between the two lenses (see Supplementary Fig. S3 for modulation transfer function and Strehl ratio simulation results). We estimate the radius of curvature of the measured membranes to be ~ 20 mm, using mechanical simulations of the structures and the observed ~ 6 - μm distance between the center of the lens and the surface of the wafer. This would result in a Strehl ratio slightly larger than 0.95. A Strehl ratio of 0.9 (as an acceptability

criterion) corresponds to a radius of curvature ~ 15 mm. If the membrane curvature is larger than this and known a priori, the lens design can be optimized to include the effects of the bending.”

3) The authors used 530 nm and 615 nm alpha-Si layer thicknesses respectively for movable and stationary lenses. Why they are chosen differently and are there any design considerations here?

Our response: The α -Si layer thicknesses (as well as the lattice constant) were chosen to maximize the efficiency of the designed high-NA lenses. The different refractive indices and thicknesses of the SiN_x membrane and the fused silica substrate result in different optimal thicknesses for the α -Si layers. Here, the choices are based on a recent algorithm developed for the design of high efficiency high-NA metasurface lenses (presented in Ref. 65). No claims are made that these are the optimal parameters resulting in the highest possible efficiencies, and more advanced optimization processes might increase these efficiencies significantly. The design procedure is explained in Ref. 65 in more detail, and a summary is given in the last few lines of page 4 and first lines of page 5:

“Since both the moving and stationary metasurface lenses have high numerical apertures ($\text{NA} \sim 0.8$), we used a recently developed technique for choosing the metasurface parameters (i.e., amorphous silicon layer thickness, lattice constant, and minimum and maximum post side lengths) to maximize the efficiency of high NA lenses for both transverse electric (TE) and transverse magnetic (TM) polarizations [65]. The method is based on approximating the efficiency of a lens designed with certain metasurface parameters through efficiencies of periodic gratings designed with the same parameters. Using this method and considering the design wavelength of 915 nm, the α -Si layer thicknesses were chosen to be 530 nm and 615 nm for the moving and stationary lenses, respectively. The lattice constant was set to 320 nm in both cases.”

4) The movable lens chip and stationary lens chip are aligned and bonded using UV-epoxy. What is the alignment accuracy here?

Our response: As discussed in Methods, a photolithography aligner (MA6/BA6 aligner, Suss MicroTec) was used to align and bond the two chips. Achieving an alignment precision of about a couple of microns is feasible with the used aligner. Using high-tech industry-level aligner/bonders, even finer alignment precision should be achievable. We have added the following to the methods section to clarify this:

“Using this technique, an alignment precision of a few microns is feasible.”

When scaling down a thin lens pair to micrometre scale, the required axial movement to achieve a desirable EFL variation ratio is drastically reduced and so is the lens aperture. The reduction in lens aperture results in an increased performance sensitivity to misalignment errors. It is expected that slight misalignment may induce a significant reduction in imaging quality. It is suggested that the authors analyse and discuss such effects.

Our response: To analyze this effect we have added MTF and Strehl ratio simulation results to Supplementary Fig. S3b, and have added the following to the discussion part to address this comment: “In addition, to have a Strehl ratio better than 0.9, the misalignment between the two lenses should be better than 2 μm . Based on the symmetric measured focal spots, we estimate the misalignment in the doublets to be smaller than this limit. Considering the high alignment precision achievable with industrial aligners, achieving a 2- μm resolution is not a challenge.”

5) On page 6 line 140, the authors measured “the focusing is close to the diffraction limit (with the measured Airy disk radius smaller than 1.1 times the theoretical value). The slight aberrations are caused by the mechanical deformation of the moving lens resulting from the residual stress in the SiNx layer.” I do not entirely agree here. When scaling down an optical system, the geometric aberration scales down significantly as well, however diffraction-limited Airy radius does not change. The authors measured 1.1 times bigger spot than the diffraction limit, which indicates the aberrations of the lens are not slight indeed.

Our response: We have changed the mentioned statements in the following way to address this comment:

“As seen in Figs. 3c and 3d, the measured Airy disk radii are smaller than 1.1 times their corresponding theoretical values. The observed aberrations ...”

6) The efficiency of the reported doublet is not high even from theoretical estimations. Is there any way to enhance this efficiency through design? For conventional lenses, efficiency can be enhanced through anti-reflection coatings, how about such meta-surface flat lenses? Please discuss in the paper.

Our response: As mentioned in the last paragraph of page 6 in the manuscript, with the current design methods, the focusing efficiency of a high-NA (~ 0.8) lens is limited to about 75%-80% (Ref. 65). This limits the efficiency of the doublet to about 55%-65%, even if the substrate reflection losses are reduced with anti-reflection coatings. However, in the past few years, there has been significant increase in the efficiency of high-NA metasurface lenses and beam deflectors. It is therefore foreseeable that the advancements in design and optimization techniques can increase these efficiencies to well above 90%. We have added the following to the first few lines of page 6:

“It is foreseeable that the efficiency can be significantly improved with better optimization and design processes [43], use of anti-reflection coatings to reduce reflection losses, and optimizing the fabrication process.”

7) On page 7 line 168, the authors stated that if the damping is reduced 20 times, the response of the device can reach 4kHz. Please briefly discuss how to control damping accurately. As far as I know, low pressure or vacuum packaging of MEMS devices usually results in oscillations and increased settling time.

Our response: As observed in Fig. 3e, the fit to the actual frequency response shows a highly overdamped system, with a damping ratio ($b/2\sqrt{mk}$) of about 10. In the mass and area scale of the membranes (i.e., mass <0.3 ng, area ~ 0.2 mm²) the dominating loss effect in atmospheric pressure is air friction [66]. Reducing the damping factor by 20, results in a slightly underdamped system, with a quality factor of about 1 (damping ratio of 0.5). This would still be a low quality factor, and the losses would still be dominated by air friction. Therefore, it should be possible to control damping by controlling the package pressure. Besides, the resonator will not have oscillation and long settling time problems with a quality factor of 1. To clarify this, we have added the following to the frequency response discussion on page 7:

“This corresponds to a highly overdamped system with a damping ratio ($b/2\sqrt{mk}$) of ~ 10 . Under the atmosphere pressure the dominant loss mechanism is the air damping [66]. If the damping is reduced by about 20 times by reducing the air pressure inside the lens packaging (i.e., $\frac{b}{2\sqrt{mk}} \approx 0.5$), then the frequency response will follow the blue dashed line in Fig. 3e, with a 3-dB bandwidth reaching 4~kHz. This would correspond to a quality factor of ~ 1 for the mechanical resonator, which should be feasible by reducing air damping. In addition, at such a low quality factor, oscillation and long settling times should not be an issue.”

In addition, for optical imaging applications such low pressure packaging seems not possible because the focal length is short here and object plane is quite close to the lens surface, which makes vacuum packing challenging.

Our response: With the current design, the focal point is about 200 μm outside the glass substrate, which is close to the working distance for some commercial objective lenses. Moreover, this distance can easily be increased to more than 500 μm by using a thinner glass substrate for the stationary lens. In addition, the silicon chip with the membrane is sandwiched between two glass substrates (one with the stationary lens, and another one attached on the backside). Aligned bonding is only required for the glass substrate carrying the stationary lens. Vacuum packaging should be relatively straightforward if the backside glass substrate is bonded to the silicon chip in a vacuum chamber with a controllable pressure. To clarify this, we have added the following to the frequency response discussion on page 7:

“Vacuum packaging could be done through bonding the backside glass substrate (the one with no metasurface) and the silicon chip carrying the membrane in a vacuum chamber with controllable pressure.”

8) The description of the triplet microscope is unclear. Where is the system aperture located? And how large the aperture is? I guess it is located at the last surface of the system (the same plane as the third meta-surface lens), but this must be clearly stated.

Our response: The design details for the triplet are given in Table S2 of the Supplementary materials. To address the comment and make the triplet data more clear, we have added the following to the triplet discussion on page 8:

“The structure is schematically shown in Fig. 5a, and is a metasurface triplet composed of a tunable doublet (with an optical design different from the fabricated one), and an additional metasurface lens. The lenses, from left to right are 540 μm , 560 μm , and 400 μm in diameter. The glass substrate is 1-mm thick, and the image plane is located 14 mm behind the third lens. The stop aperture is located at the plane of the right-most lens, and has the same diameter of 400 μm (see Supplementary Fig. S1 for phase profiles, and Table S2 for the corresponding coefficients).”

In addition, why such a triplet system has a low numerical aperture of only 0.16, while the previous descriptions of the doublet lenses clearly indicate the numerical aperture around 0.8 for the doublets? I guess this is because of the optical design? Please discuss. And, why the diffraction limited Airy disks have about 40-micron radii, this is quite large even for 910 nm wavelength light?

Oure response: Each of the elements in the fabricated doublet have numerical apertures of about 0.8 (diameters of 300 μm , and focal distances of 119 μm and -132 μm). However, as seen in Fig. 3a, the effective focal length of the doublet changes from 350 μm to about 700 μm . For values closer to the

optimal operation regime (e.g., EFL \sim 600 μm), the numerical aperture of the doublet is about 0.25. This is actually a key requirement in the design that each element has an optical power significantly higher than the combined optical power. This enables the high ratio of change in the focal distance to the movement of the membrane (i.e., a ratio of >30 to 1). The triplet has a different design, with the membrane and first glass lenses having diameters of ~ 550 μm , and numerical apertures of ~ 0.7 (focal distances of -290 μm and 275 μm). The second glass lens is much weaker, with a focal distance of ~ 1500 μm . In this case too, the total optical power is about four times smaller than each of the elements in the tunable doublet (EFL \sim 1200 μm). Given the aperture of ~ 370 μm (resulting from the small positive power of the second glass lens), an object space numerical aperture of 0.16 results. Using the same aperture, and the 14 mm distance from the aperture to the image plane, an image plane numerical aperture of about 0.014 is calculated. This results in a diffraction limited Airy radius of ~ 40 μm in the image plane. On the other hand, the imaging resolution (given by the object space NA of ~ 0.16) is about 3.5 μm . To clarify these, we have added the following to the description of the triplet:

“The effective focal length for the whole system is ~ 1160 μm (for $d=9$ μm case), which is significantly larger than the focal lengths of the membrane and first glass lenses, similar to the fabricated doublet. As a result, the object space NA is about 0.16, corresponding to an imaging resolution of ~ 3.5 μm . Considering the 14-mm distance between the image plane and the backside aperture, the image space NA is ~ 0.014 which results in an Airy radius of about 40 μm in the image plane.”

We have also added the following to page 4, to provide more information about the fabricated doublet:

“The membrane and glass lenses are 300 μm in diameter, and have focal lengths of 120 μm and -130 μm , respectively.”

Overall, this paper can be recommended to be accepted for publication only when all the above issues are cleared.

Reviewers' comments:

Reviewer #1 (Remarks to the Author):

This paper presents discussion and demonstration of a varifocal lens system utilizing MEMS technology to control the longitudinal distance between two thin lens profiles implemented as dielectric metasurfaces. Control of the distance between the metasurfaces controls the overall optical power of the system. Very basic discussion of the operation of the system is presented, along with more detailed discussion of fabrication and testing/characterization. The paper is well written with good figures and detail in the supplementary documents. This work is likely to be of interest to others in the field of metasurfaces, MEMS, and imaging systems.

I note that the authors have already demonstrated the use of metasurface lens doublets for imaging systems, and other researchers have demonstrated the uses of MEMS and metasurfaces to vary optical functions. With this in mind, the work in this paper is of very high quality, but not necessarily novel in the context of this prior work.

Reviewer #3 (Remarks to the Author):

The authors have revised the manuscript according to my comments, but there are still issues that are not entirely resolved. These are listed below:

1) Not satisfactory. I do not agree the importance of the large absolute optical power tunability described by the authors. The authors used the large tunability of objective distances as an example. This is not sufficient and also not a good support to this claim. In my view, any lens with any small tunability of optical power can achieve a large objective distance tunability as long as the condition is right. My reason is as follows: take the simple lens equation $1/L_o + 1/L_i = 1/f$, where L_o , L_i , and f are respectively the object distance, image distance, and focal length. If the focal length changes to $f+D$, the equation becomes $1/L_o' + 1/L_i = 1/(f+D)$, assuming the image plane is fixed. It is quite easy to obtain the ratio (L_o'/L_o) is approximately equal to $(D/f)(L_o'/f) + 1$, if the change of focal length D is much smaller than f (i.e. $f \gg D$). It can be seen that if (L_o'/f) is sufficiently large, one can achieve any tunability ratio of the object distance (L_o'/L_o) with small (D/f) as long as (L_o'/f) is large. In view of this, the importance of the design is not quite justified.

2) Satisfactory.

3) Satisfactory.

4) Okey.

5) Okey.

6) Okey.

7) Satisfactory, to some extent. Controlling precisely the pressure in MEMS packaging is challenging.

8) Satisfactory to some extent. I am still not convinced why the overall number aperture is low here. Is there any fundamental limitation in this microscope design? The 3.5-micron object resolution is not very useful in many biological applications. By the way, it should read instead "... corresponding to a resolution of $\sim 3.5 \mu\text{m}$ at the object plane." to avoid the confusion.

Second round of reviews:

Reviewers' comments:

1) Not satisfactory. I do not agree the importance of the large absolute optical power tunability described by the authors. The authors used the large tunability of objective distances as an example. This is not sufficient and also not a good support to this claim. In my view, any lens with any small tunability of optical power can achieve a large objective distance tunability as long as the condition is right. My reason is as follows: take the simple lens equation $1/L_o + 1/L_i = 1/f$, where L_o , L_i , and f are respectively the object distance, image distance, and focal length. If the focal length changes to $f+D$, the equation becomes $1/L_o' + 1/L_i = 1/(f+D)$, assuming the image plane is fixed. It is quite easy to obtain the ratio (L_o'/L_o) is approximately equal to $(D/f)(L_o'/f) + 1$, if the change of focal length D is much smaller than f (i.e. $f \gg D$). It can be seen that if (L_o'/f) is sufficiently large, one can achieve any tunability ratio of the object distance (L_o'/L_o) with small (D/f) as long as (L_o'/f) is large. In view of this, the importance of the design is not quite justified.

Our response: We are sorry that the wording of our previous response has generated this misunderstanding. The reviewer is correct in mentioning that even a small change in the focal length can result in a large change in the object distance under the right conditions. However, a large change in the object distance is not the main support for our claim. Our claim is that when the tunable MEMS doublet is integrated into an optical system with an overall optical power that is small compared to the optical power of the MEMS doublet, the overall *effective focal length* of the system can be tuned substantially. In the case of the system in Fig. 4, it is the *effective focal length* that changes from 44 mm to 122 mm (i.e., a change of about 2.8 times). This is different from the case that the reviewer discusses where a small change in the focal length can change the object distance significantly. In other words, our claim is that the EFL itself can also be substantially tuned using the doublet. To Make this clear, we have reworded the corresponding paragraph on page 8:

“As observed here, by moving the membrane only about 4 μm , the overall system EFL changes from 44 mm to 122 mm, a ratio of about 1:2.8. This is an example of the importance of the large absolute optical

power tunability of the metasurface doublet, especially when it is integrated into a system with a comparably small overall optical power. It also allows for changing the object distance from 4 mm to 15 mm by electrically controlled refocusing.”

7) Satisfactory, to some extent. Controlling precisely the pressure in MEMS packaging is challenging. Our response: This is correct, however, in the case of the MEMS doublet the pressure does not need to be precisely controlled. The pressure should only be low enough to avoid a highly overdamped system (i.e., $b < \sqrt{mk}$).

8) Satisfactory to some extent. I am still not convinced why the overall number aperture is low here. Is there any fundamental limitation in this microscope design? The 3.5-micron object resolution is not very useful in many biological applications. By the way, it should read instead “... corresponding to a resolution of $\sim 3.5 \mu\text{m}$ at the object plane.” to avoid the confusion.

Our response: We have changed the text to “corresponding to a resolution of $\sim 3.5 \mu\text{m}$ at the object plane.” to reflect the reviewer’s comment. The ~ 0.16 numerical aperture is simply a result of the chosen lens aperture and overall focal distance and tunability. We chose these parameters because of our interest to apply these devices to neural microscopy where we only want to image the cell bodies (dimension of a few microns). The NA can be increased or decreased by changing the design parameters. However, a more in-depth study of the ultimate limits of the platform would be the subject of a further study and outside the scope of the current manuscript.